# An Integrated LSTM-Rule-Based Fusion Method for the Localization of Intelligent Vehicles in a Complex Environment

**DOI:** 10.3390/s24124025

**Published:** 2024-06-20

**Authors:** Quan Yuan, Fuwu Yan, Zhishuai Yin, Chen Lv, Jie Hu, Yue Li, Jinhai Wang

**Affiliations:** 1Hubei Key Laboratory of Advanced Technology for Automotive Components, Wuhan University of Technology, Wuhan 430070, China; 231943@whut.edu.cn (Q.Y.); wangjinhai@whut.edu.cn (J.W.); 2Hubei Collaborative Innovation Center for Automotive Components Technology, Wuhan University of Technology, Wuhan 430070, China; zyin@whut.edu.cn (Z.Y.); auto_hj@163.com (J.H.); 3The Automated Driving and Human-Machine System Group, Nanyang Technological University, Singapore 639798, Singapore; lyuchen@ntu.sg.edu; 4Dongfeng Technology Center, Wuhan 430056, China; yx-liy@dfmc.com.cn

**Keywords:** multi-source fusion, fuzzy rules, trajectory matching, dual-LSTM

## Abstract

To improve the accuracy and robustness of autonomous vehicle localization in a complex environment, this paper proposes a multi-source fusion localization method that integrates GPS, laser SLAM, and an odometer model. Firstly, fuzzy rules are constructed to accurately analyze the in-vehicle localization deviation and confidence factor to improve the initial fusion localization accuracy. Then, an odometer model for obtaining the projected localization trajectory is constructed. Considering the high accuracy of the odometer’s projected trajectory within a short distance, we used the shape of the projected localization trajectory to inhibit the initial fusion localization noise and used trajectory matching to obtain an accurate localization. Finally, the Dual-LSTM network is constructed to predict the localization and build an electronic fence to guarantee the safety of the vehicle while also guaranteeing the updating of short-distance localization information of the vehicle when the above-mentioned fusion localization is unreliable. Under the limited arithmetic condition of the vehicle platform, accurate and reliable localization is realized in a complex environment. The proposed method was verified by long-time operation on the real vehicle platform, and compared with the EKF fusion localization method, the average root mean square error of localization was reduced by 66%, reaching centimeter-level localization accuracy.

## 1. Introduction

Autonomous driving has been widely believed to have great potential for the improvement of road safety, energy efficiency, and traffic congestion [1,2]. In recent years, autonomous vehicles are gradually moving toward large-scale open road applications [3]. Faced with complex driving environments, higher requirements are put forward for path planning and trajectory tracking [4]. Accurate and reliable localization is the key basis for autonomous vehicles to accomplish safe path planning and achieve accurate trajectory tracking [5]. SLAM (Simultaneous Localization and Mapping) and GPS (Global Positioning System) are commonly used for vehicle localization [6,7]. However, in complex and variable environments, the blocking and interference of vehicle localization signals are widespread, which greatly reduces the localization accuracy and reliability of GPS and SLAM [8,9]. In summary, achieving accurate and reliable localization in complex and variable environments is one of the urgent problems to be solved for autonomous vehicles [10].

To achieve better accuracy localization in complex and variable environments, fusing multi-source information has become a popular choice [11]. Generally, these localization systems can be summarized as a combination of computational algorithms with multi-source inputs. For GPS-denied environments, many methods have been proposed to guarantee localization accuracy. Considering the superiority of the low cost of the camera sensor, Ref. [12] proposed a novel localization method with prior dense visual point cloud map constraints generated by a stereo camera, Ref. [5] combined both INS and Vision to provide robust and continuous navigation output in complex driving conditions, Ref. [13] fused visual-inertial odometry (VIO) measurements with wheel odometry data using an extended Kalman filter (EKF), and Ref. [14] proposed a robust localization system combining vision and lidar. The approaches mentioned can be categorized as using in-vehicle sensors, while others tend to take advantage of smart cities to improve localization accuracy. Ref. [15] made use of a wireless sensor network, according to the signal intensity, to deduce a position. The vehicle-to-everything technology can also be applied to improve localization accuracy in severe environments [16,17,18]. However, building a smart city faces many challenges. Firstly, a large site is required, in addition to the cost of developing regulations and building to operate autonomous vehicles in the smart city, making it difficult to realize. High-Definition (HD) maps seem to be a more acceptable choice; the proper use of HD maps can improve the accuracy of localization [19]. However, the cost of HD map information collection for vehicles is quite high, and the work of analyzing and processing the information is massive.

In addition to the multi-inputs to be applied to localization, many algorithms have been applied. Widely used in autonomous vehicles are Particle Filter (PF) algorithms. Rao-Blackwellized Particle Filters (RBPFs) were proposed, which use a preprocessing stage to detect vertical objects in the original scan [20]. Others presented a modified PF weight updating algorithm for precise in-lane localization by fusing information from visual lane markers and GPS [21]. Nowadays, researchers focus on reducing the computational cost of algorithms. Thus, a map-sharing strategy was proposed, in which the particles only have a small map of the nearby environment [22]. With the improvement in the computation capacity of in-vehicle hardware, machine-learning algorithms have been widely applied in localization [23]. In [24], the author introduced a location-based car park system based on the conventional neural network. This system was used to localize and identify the cars in the parking area. Contemporary localization integrated with machine-learning algorithms uses different kinds of input data, such as inertial sensor data [25], camera data [26], sound data [27], and lidar (light detection and ranging) [28]. In addition to neural network applications, Reinforcement Learning (RL) is used to produce a policy distribution that maps system states to action sets to maximize reward. RL can improve the accuracy of GPS through the ‘correction’ operation of an inexpensive GPS unit. The proposed model does not require any auxiliary equipment nor does it make rigid assumptions about the noise parameters of the GPS device or the motion models [29]. Ref. [30] used Deep Reinforcement Learning (DRL), which provides an elegant mapless approach that integrates image processing, localization, and planning in one module, reaching a 0.3 m neighborhood of the goal in more than 86.7% of cases. However, high computation capacity hardware also means a high cost.

All the solutions can be summarized as solving the problem of localization in GPS-absent environments, but in reality, the question still remains: what do we do if the environmental features are obscured by obstacles? In [11], the author explored the influence of the deep urbanized city on the accuracy of NDT (Normal Distributions Transform)-based graph SLAM. The accuracy of the SLAM is degraded with increased traffic density. Considering that 2D lidar has the advantage of low cost and high accuracy, Refs. [31,32] used deep Recurrent Convolutional Neural Networks (RCNNs) for odometry estimation using only 2D laser scanners. However, the proposed method has a complex architecture and high computing power requirement, which may affect other processes in the case of limited in-vehicle computing power. Many researchers have tried to extract different features from maps to improve the localization accuracy of SLAM methods, such as lines, curves, etc. [33]. However, these methods have a high time and energy cost, and the localization accuracy decreases rapidly in the above feature-obscuring scenarios. In addition to laser SLAM, visual SLAM has also been widely studied because of its significant cost advantages [34]. A fusion of a 2D image with IMU or depth data greatly improves localization accuracy [35]. In complex and variable scenes, occlusion is widespread, and exposure at the intersection of unobscured and occluded scenes may occur, which has a large impact on vision camera applications while dynamic obstacles may also occlude scene localization features. The combination of depth data also means the high computing pressure on the platform.

In addition to what is mentioned above, to guarantee the autonomous vehicle application, cost control is a problem that we cannot ignore [9]. According to the challenge described above, this paper introduces a scheme method for severe environment localization named the Rule-Based-Fusion-LSTM module. Firstly, we formulated a series of rules to define the confidence factor of the GPS and SLAM localization and acquire the initial fusion localization. Then, we constructed an odometer model for obtaining the projected localization trajectory. Considering the high accuracy of the odometer’s projected trajectory within a short distance, we used the shape of the projected localization trajectory to inhibit the initial fusion localization noise and used trajectory matching to obtain an accurate localization. Finally, the Dual-LSTM network is constructed to predict the localization and build an electronic fence to guarantee the safety of the vehicle while also guaranteeing the updating of the short-distance localization information of the vehicle when the above-mentioned fusion localization is unreliable.

This study produced three primary contributions: (1) An accurate and reliable localization method in a complex environment is proposed by combining direct localization, reckoning localization, and trajectory prediction effectively. (2) We proposed a method of accurately judging the real state of GPS and SLAM localization and combined them. (3) We guaranteed the computational capacity of the vehicle controller can satisfy the localization deviation.

The remainder of this paper is organized as follows. Section 2 illustrates the scheme of this paper. In Section 3, the results and the contrast to the related methods are concluded. Conclusions are offered in Section 4.

## 2. Approach

### 2.1. System Architecture

In this paper, the proposed system architecture is presented in Figure 1. It mainly consists of three modules, the fuzzy fusion-based localization module, the odometer movement track matching module, and the trajectory prediction module. From a real operating autonomous vehicle, we find in a complex environment that the vehicle localization deviation does not match the localization system outputs. Thus, we designed a rule-based module to define the confidence factor of GPS and SLAM localization. Considering SLAM has the LoopClosure problem and the localization performance will decrease as the dynamic obstacles around the vehicle increase, combining the SLAM and GPS acquires the initial fusion localization. However, in a complex environment, the initial fused localization will drift, and considering the high accuracy of the odometer’s projected trajectory within a short distance, we used the shape of the projected localization trajectory to inhibit the initial fusion localization noise and used trajectory matching to obtain an accurate localization. When the GPS and SLAM localization drifts for more than a period of time, localization is obtained by combining the optimal rotational translation matrix at the previous moment with the odometer-fitted trajectory. When the above fusion localization is unreliable, we used the Dual-LSTM network to predict the vehicle trajectory according to the historical data, using the prediction trajectory to guide the vehicle. In the meantime, we used the lidar sensor to build the electric fence to guarantee vehicle safety. Compared to existing studies, this architecture does not add the extra sensor, making full use of the low-cost, in-vehicle sensor to solve the localization problem and define the confidence factor of both localization methods to avoid deviation. Using the relatively simple NN rather than RL, we ensure the computational capacity of the in-vehicle controller can be satisfied.

### 2.2. Fuzzy Fusion-Based Localization Module

The common fusing methods are based on the assumption that the state data from the module are reliable. However, during the vehicle’s autonomous operation, we find in a complex environment that the vehicle localization deviation does not match the localization outputs. In Figure 2, it shows the GPS states in the complex environment. At the first sharp jump, the gps_error was suddenly increased to over 1 m, and then it declined to 0.16 m. The Flag_Pos was suddenly decreased to 17 and then gently increased to 34. According to the GPS manual, the accuracy of the GPS localization was enough for vehicle navigation. However, the real vehicle performance was not corresponding to it. The vehicle deviated from the route and exited the safe boundary area. The performance of SLAM is similar to GPS, especially in scenarios with lots of dynamic obstacles. Based on this, we designed the fuzzy fusion-based localization module to define the confidence factor of the GPS and SLAM localization outputs and then acquire the initial fusion localization.

The fuzzy fusion-based localization module has two localization inputs, GPS and SLAM. The structure of the GPS module is relatively simple. For the SLAM module, we use NDT to compute point cloud interframe transformations. Point Cloud Library is a large open-source project for 2D/3D image and point cloud processing. The PCL framework consists of many advanced algorithms, including filtering, feature estimation, surface reconstruction, registration, model assembly, and segmentation [36]. This paper uses IMU-based odometry to give the rough position and publish the exact position by a non-destructive update implemented in PCL.

The odometry results were passed to the NDT algorithms as the rough starting position; the NDT algorithm then used PCL to obtain the accurate position. The NDT innovatively divides the point cloud space into cells (each cell is continuously modeled by a Gaussian distribution). In this case, the discrete point clouds are transformed into successive continuous functions [11]. The process of calculating the relative pose between the reference and the input point clouds is listed as follows:

(1) Normal distribution transform: fetch all the points, $x_{i=1\ldots n}$, contained in 3D in the cell. Calculate the mean among all the points: $q=\frac{1}{n}\sum_{i}x_{i}$. Calculate the covariance matrix μ:(1)$$\mu =\frac{1}{n}\sum_{i}(x_{i}-q){\left(x_{i}-q\right)}^{T}.$$

(2) The matching score is modeled as
(2)$$f(p)=-score(p)=\sum_{i}\exp(-\frac{{\left({x^{\prime }}_{i}-q_{i}\right)}^{T}u_{i}^{-1}({x^{\prime }}_{i}-q_{i})}{2}),$$
where $x_{i}$ represents the points in the current frame of the lidar scan, ${x^{\prime }}_{i}$ represents the points in the previous scan mapped in the current frame using the transfer matrix T ($T=\left[r_{x}\;r_{y}\;r_{z}{\;o}_{x}{\;o}_{y}{\;o}_{z}\right]$) $q_{i}$, and $u_{i}$ means the mean and covariance of the corresponding normal distribution, respectively.

(3) Update the pose using the Newton method to minimize the score of f(p).

(4) Repeat Steps 2 and 3 until maximizing f(p).

After achieving the GPS and SLAM localization, the proposed method does not just believe the output of the localization system module, it observes the state messages of the localization system module over a period of time, giving the confidence factor of these states at time t1, and then based on the defined rule, combining the confidence factor and state at t1 to fuse the multi-input position. In addition to the rule-based module mentioned above, the fusing module also uses the neural network to re-adjust the fusing weights. The first layer of the neural network was trained to recognize the vehicle motion, transfer the position data to NN separately, and compare the output motion with the real history trajectory, which can improve the confidence of the sensor module output reliability.

For the input timing length, ten cycle time lengths were chosen as the trajectory data input length transfer to the confidence define module. For the GPS confidence definition, gps_error, Flag_Pos, and NumSV, these states were taken into consideration, and the confidence factor of GPS is calculated as follows:(3)$$\begin{array}{cc} D_{GPS} & =w_{1}(0.1error_{0}+0.2error_{1}+\ldots +0.9error_{8}+error_{9})\\ & +w_{2}(0.1\times FlagPos_{0}/50+\ldots +FlagPos_{9}/50)\\ & +w_{3}(0.1\times NumSV_{0}/30+\ldots +NumSV_{9}/30) \end{array}$$
where the error is the gps_error, FalgPos is the GPS position state signal, NumSV means the number of satellites received, and w1, w2, and w3 are 0.4, 0.3, and 0.3, respectively. Considering the influence of historical trajectory data on vehicle localization, the closer the time distance is, the larger the factor weights are.

For the SLAM module, the states of fitness_score and iteration were considered. The confidence factor for SLAM is calculated as follows:(4)$$\begin{array}{cc} D_{SLAM} & =w_{1}(0.1score_{0}+0.2score_{1}+\ldots +0.9score_{8}+score_{9})\\ & +w_{2}(0.1(10-itea_{0})÷10+\ldots +(10-itea_{9})÷10) \end{array}$$
where the score is fitness_score, which is the score obtained during the NDT lidar scheme matching. The itea represents the iteration for the NDT algorithm experienced to match the lidar scheme and w1 and w2 are 0.7 and 0.3.

After obtaining the output confidence factor of the GPS and SLAM localization system, the states and confidence factor are transferred to the fuzzy rule. Based on fuzzy modeling rules, and combining GPS and NDT system localization information and confidence factors, we acquire the initial fusion localization results. The output of the GPS localization system and its confidence factor, and the output of the SLAM localization system and its confidence factor, are fuzzy control inputs, and the output is the weight factor W1.

The input variables are fuzzy processed, and the fuzzy rules are divided into seven fuzzy subsets, which are, respectively, Strong Negative, Negative, Little Negative, Zero, and Little Positive. The output variable is fuzzy and the fuzzy rule is divided into five fuzzy subsets, which are Negative, Zero, Little Positive, Positive, and Strong Positive, respectively. A Gaussian-type function is selected as the membership function. Part of the defined rules is shown in Table 1.

The output parameter W1 is the weight coefficient of the GPS localization output in the fusion localization, which is divided into five levels (Strong Positive, Positive, Little Positive, Zero, and Negative), where a negative value of W1 indicates that both the GPS and SLAM localization have drifted at this time, and the unreliable marker of localization status is transferred to the odometer motion trajectory matching system, and the current vehicle localization is obtained by combining the calculated localization and the optimized rotation and translation matrix of the previous time.

After the weight factor W1 is obtained based on the fuzzy rules, the GPS and NDT localization output localization information is transmitted to the first layer of the LSTM network in the trained Dual-LSTM network, and the output result of the neural network is compared with the real trajectory information. The weight factor W1 is dynamically adjusted according to the prediction errors GPS_pre_Error and NDT_pre_Error. When the prediction error result deviates greatly from the weight factor situation, the weight factor W1 is corrected in time, as shown in the following equation.
(5)$$W1_{new}=W1*\frac{NDT\_pre\_Error}{GPS\_pre\_Error}$$

According to the weight factor W1 after correction, GPS and NDT localization are integrated, and the equation is shown as follows:(6)Position=W1∗GPS_position+(1−W1)∗NDT_position.

### 2.3. Odometer Movement Track Matching Module

This paper uses IMU-based odometry to obtain the projected localization trajectories, which was shown as Figure 3. The yellow dot in the figure indicates that the center of mass of the vehicle model is the center of the rear axle. The IMU-based odometry is calculated from the following formula:
(7)$$\left\{\begin{array}{l} \Delta \theta =\Delta s_{r}-\Delta s_{l}/L\\ r=\Delta s_{r}+\Delta s_{l}/2*\Delta \theta \\ \Delta x=r\sin(\Delta \theta )\\ \Delta y=r(1-\cos(\Delta \theta ))\\ \theta_{1}=\theta_{0}+\Delta \theta \\ x_{1}=x_{0}+\Delta x\cos(\theta_{0})-\Delta y\sin(\theta_{0})\\ y_{1}=y_{0}+\Delta y\sin(\theta_{0})-\Delta y\cos(\theta_{0}) \end{array}\right.$$
where $\Delta s_{r}$ is the distance the right wheel passed, $\Delta s_{l}$ is the distance the left wheel passed, r is the vehicle turning radius, and L means the wheelbase.

The error of the odometer model will increase with time, but within a certain range over a short period of time, it can be assumed that the calculated localization result is very close to the actual localization. We separately compute the localization results calculated by the odometry model and the direct output of the fused localization for a certain period of time. In the formula, n represents the consideration of the track point information. It can be considered that when n is within a certain range, the track is almost consistent; thus, you only need to find the optimal rotation–translation matrix to achieve full fitting. In other words, if you find the optimal rotation–translation matrix, you can obtain accurate global localization information of the vehicle through the odometer calculation results. Its principle is shown in Figure 4 below.

We believe that there is a rotation angle and a corresponding translation deviation between the direct localization result and the recommended localization result, so as to construct a rotation–translation matrix and convert the odometer trajectory to the global path.
(8)$$R_{m}=\left[\begin{array}{l} \begin{array}{cc} \cos\theta_{m} & -\sin\theta_{m} \end{array}\\ \begin{array}{cc} \sin\theta_{m} & \cos\theta_{m} \end{array} \end{array}\right]$$
(9)$$B_{m}=\left[\begin{array}{l} \Delta x_{m}\\ \Delta y_{m} \end{array}\right]$$
(10)$$T_{k}^{\prime }=T_{k}R_{k}+B_{k}$$

Then, calculate the deviation between $T_{k}^{\prime }$ and $G_{k}$, and use the optimization solution to obtain the optimal rotation and translation matrix. $G_{k}$ is the fusion localization in part B.
(11)$$J(P)=\left\{G_{k}-T_{k}^{\prime }|k=1,2,\ldots ,n\right\}$$

In the formula, n represents the consideration of the trajectory length.

This chapter uses the particle swarm optimization algorithm to solve the optimization problem. Its formula is defined as follows:(12)$$v_{ij}(t+1)=w*v_{ij}(t)+c_{1}r_{1}(t)[p_{ij}(t)-x_{ij}(t)]+c_{2}r_{2}(t)[p_{gj}(t)-x_{ij}(t)]$$
(13)$$x_{ij}(t+1)=x_{ij}(t)+v_{ij}(t+1)$$
(14)$$w=w_{\max}-\frac{(w_{\max}-w_{\min})*t}{T_{\max}}.$$

The rotation and translation matrix that minimizes the function can be obtained by optimizing the solution. After obtaining the optimal rotation and translation matrix, the accurate global trajectory shape can be obtained by fitting the trajectory and the optimal rotation and translation matrix with the odometer. The shape can be used to inhibit the initial fusion localization noise and obtain the accurate localization result of the vehicle. When the GPS and NDT localization drift, the vehicle localization results can be obtained in a certain time domain by combining the odometer to calculate the trajectory and the optimal rotation and translation matrix at the above time, so as to ensure the safe and reliable driving of the autonomous vehicle.

### 2.4. Trajectory Prediction Module

When a vehicle is traveling in a scenario with GPS signals that are blocked and there exist many dynamic obstacles over a period of time, the above-mentioned localization still drifts. Thus, we use the Dual-LSTM network to predict the vehicle trajectory according to the historical data, using the prediction trajectory to guide the vehicle. In the meantime, we use the lidar sensor building the electric fence to guarantee vehicle safety. Compared with the normal LSTM network, Dual-LSTM uses the first LSTM layer to predict the vehicle motion and then combines the historical trajectory information with the motion using the second LSTM layer outputs of the predicted trajectory, which has higher accuracy. In addition to this, this paper uses the first LSTM layer to dynamically adjust the initial fusion localization weights.

Based on what is mentioned above, the main architecture of the trajectory prediction module can be summarized as the Dual-LSTM scheme. In this paper, the predicted trajectory length is set at 5 s; for the length chosen, too short is not enough for vehicles passing the scene so that the environment features are obstructed by the obstacle, too long will increase the challenge of keeping the error small enough for vehicle navigation. In order to balance these two contradictions, and meanwhile considering the vehicle operation data, 5 s is suitable. For the seq2seq problem, this paper takes the LSTM encode–decode mode, which is shown as Figure 5. The LSTM encoder–decoder consists of two LSTMs. The first LSTM (encoder) processes an input sequence and generates an encoded state. The encoded state summarizes the information in the input sequence. The second LSTM (decoder) uses the encoded state to produce an output sequence. The encode–decode mode can be shown as follows.

In this architecture, the first LSTM layer aims to predict the vehicle motion, e.g., acceleration or deceleration, and transform the features of the sequential input combined with the trajectory data to the second LSTM layer for the prediction of the vehicle position. The data processing module windowed the training data. The efficacy of it is explained in Figure 6.

The LSTM has three gates (forget gate, input gate, and output gate) and a memory cell, shown as Figure 7. For the LSTM block, this can be summarized as three steps. First, the forget gate controls the sigmoid function, determining which data pass the cell state. Second, the input gate using the sigmoid function defines which data to be updated, and the tanh layer produces the new candidate values. The third step is choosing the module outputs. The equations are shown as follows:(15)$$i_{t}=\sigma (w_{x^{i}}x_{t}+w_{h^{i}}h_{t-1}+b_{i})$$
(16)$$f_{t}=\sigma (w_{x^{f}}x_{t}+w_{h^{f}}h_{t-1}+b_{f})$$
(17)$$O_{t}=\sigma (w_{x^{o}}x_{t}+w_{h^{o}}h_{t-1}+b_{o})$$
(18)$$g_{t}=\tanh(w_{x^{c}}x_{t}+w_{h^{c}}h_{t-1}+b_{c})$$
(19)$$c_{t}=f_{t}⊙c_{t-1}+i_{t}⊙g_{t}$$
(20)$$h_{t}=o_{t}⊙\tanh(c_{t}),$$
where $\sigma (x)=1/(1+e^{-x})$ is the sigmoid function and $i_{t}$, $f_{t}$, $O_{t}$, and $g_{t}$ are the input gate vector, forget gate vector, output gate vector, and state update vector, respectively. $w_{x^{i}}$, $w_{x^{f}}$, $w_{x^{o}}$, $w_{x^{c}}$, $w_{h^{i}}$, $w_{h^{f}}$, $w_{h^{o}}$, and $w_{h^{c}}$ are the weights for linear combination, $b_{i}$, $b_{f}$, $b_{o}$, and $b_{C}$ are the bias, and ⊙ represents the element-wise production.

In the neural network training process, the loss function was different from the operating process. The prediction time length is 5 s, at the rate of 50 Hz, and the predicted data length is 250. Considering the shorter time span, the data have more influence on vehicle navigation and vehicle decision-making. For the loss function, this is shown as Equation (21).
(21)$$Loss=\sum_{i=0}^{n}w_{i}(output_{i}-target\_batch_{i}),$$
where $w_{i}$ is the weights of the data during every prediction batch.
(22)$$\left\{\begin{array}{l} w_{i}=(length(data\_batch)-i)*2/(length(data\_batch)+1)\\ \sum_{i=0}^{n}w_{i}=n \end{array}\right.$$

## 3. Experiments

This paper’s proposed method was tested on the sharving-one 1.0 plus vehicle platform, shown as Figure 8. After being trained, the LSTM trajectory model was utilized in the ROS (Robot Operating System) as a block. To evaluate the performance of the proposed method, the trajectory prediction module was compared with the method of EKF, and the result of the whole module is shown in Section 3.1.

### 3.1. Routing Map

The autonomous vehicle operating route map is shown in Figure 9. The routing length of the vehicle test was 683 m, consisting of a variety of environmental features for algorithm verification, such as the sheltered area (501 m to 553 m and 269 m to 348 m), tall-buildings sheltered area (348 m to 436 m), covered shelter (from 201 m to 248 m), and a complex scene with tree shelters and dynamic traffic from 553 m to 683 m. The upper limit of the vehicle operating speed was set at 25 km/h.

### 3.2. Trajectory Prediction Part

The dataset was collected by the SV 1.0 plus real vehicle platform between May 2021 and October 2021. The dataset consists of GPS localization, SLAM localization, the localization acquired through the methods in this paper that combined the initial localization and the projected localization, and all the information of the autonomous vehicle transmitted to the CAN line of the vehicle chassis. The dataset contains nearly 1.2 million vehicle track points with a sampling rate of 50 Hz. After obtaining the sampled data, the vehicle’s driving intent is identified and the driving intent list label is automatically added to the dataset. The data under the occlusion and interference of localization signals were extracted from the dataset to form a complex and variable scene dataset for the test and verification of the self-vehicle trajectory prediction model. Nearly 7000 sample fragments were extracted; the first 5700 sample fragments were selected as the training set, and the last 1300 sample fragments were selected as the test set. In order to facilitate verification and comparison, the test vehicle is equipped with a higher precision integrated inertial navigation system, and its output position is recorded as the truth value information at the same time.

In the process of network training, the input window is set to 200, the stride length is set to 50, and the output window is set to 250. The historical trajectory and speed information of the past 4 s are used to predict the self-vehicle trajectory of the future 5 s. The loss function uses the above custom loss function, as shown in Equations (21)–(22). The initial learning rate is set to 0.001 and the hidden layer size is set to 128.

Figure 10 shows the Dual-LSTM network prediction error and fusion positioning error output in the verification set. The horizontal coordinate is the sample fragment count.

The test results show that in a long-term and complex environment, the maximum fusion positioning error can reach about 2.5 m, and there are vehicles running out of the driving lane and risking collision in the real vehicle operation acquisition, while the prediction error of the self-vehicle trajectory prediction model is more accurate than that of the construction of the self-vehicle trajectory prediction model, and the maximum error is no more than 0.6 m, which proves that the construction of the self-vehicle prediction model is very important. The trained self-vehicle trajectory prediction model is used as a function package, and the above functions are further tested and verified on the real vehicle.

### 3.3. Whole Module In-Vehicle Test

Next, we discuss the results for each localization method and the proposed method. All experiments ran within a single thread on the SV 1.0 plus platform. Finally, the trained LSTM network was used to predict the trajectory data based on the real-time updated data. The whole architecture was used in the autonomous vehicle operation. The results of the vehicle test are shown below. Figure 11a shows the final whole map of the proposed method. The detailed comparison results can be seen in Figure 12a,b. Figure 12a is the overall comparison of the localization methods in vehicle operation. Figure 12b shows the output fusion weight coefficient of the initial localization fusion module in the proposed method, which proves that GPS and SLAM localization can be properly fused under well localization conditions. At the same time, when the localization drift, negative output indicates that both GPS and SLAM are drifting, and the initial localization is not reliable. Then, combining the calculated localization to improve the localization accuracy and reliability.

Figure 12a shows a real vehicle positioning situation, where the red line represents the localization results of the GPS module, the blue line represents the localization results of the NDT module, the green line represents the real position of the vehicle, and the black line represents the localization outputs of the proposed method. In Figure 12a, compared with the GPS and NDT localization, the proposed method in this paper is closer to the real position and has better localization. Figure 12b shows the localization in a complex environment; it can be seen that both the GPS and NDT localization were not reliable. The localization error of GPS and NDT was more than 1 m, whereas the proposed method still performed well; the localization of the proposed method was close to the real position.

The results of GPS, NDT, and the proposed method were compared together. Figure 13a,b shows the localization error of the three methods during the in-vehicle test. From the figure, the proposed method has a smaller error; the maximum localization error of the proposed method is 0.78 m, while the GPS module and NDT SLAM module are 1.55 m and 1.23 m, respectively. The mean square errors of latitude and longitude for the proposed method were 0.08 m and 0.12 m, respectively, while the GPS module’s mean square errors of latitude and longitude were 0.28 m and 0.42 m, respectively, and the NDT module’s mean square errors of latitude and longitude were 0.19 m and 0.28 m, respectively. In addition, we have compared the proposed method with the EKF method, which fuses the signal of the GPS and IMU, and the results of the comparison are shown in Figure 14. The maximum localization X error of the proposed method is 0.54 m while it is 1.3 m for the EKF method; the maximum Y error of the proposed method is 0.77 m while it is 1.7 m for the EKF method. The mean square errors of latitude and longitude for the EKF method were 0.24 m and 0.36 m, respectively. It can be seen that the localization effect of the proposed method is better than that of the EKF method in the test environment. The detail comparison of EKF and proposed method are in Table 2.

## 4. Conclusions

This paper proposed a Rule-Based-Fusion-LSTM method that combined the rule-based module and learning-based network to deal with autonomous vehicle self-localization in complex environments. Firstly, fuzzy rules are constructed to accurately analyze the in-vehicle localization deviation and confidence factor to improve the initial fusion localization accuracy. Then, we constructed an odometer model for obtaining the projected localization trajectory. Considering the high accuracy of the odometer’s projected trajectory within a short distance, we used the shape of the projected localization trajectory to inhibit the initial fusion localization noise and used trajectory matching to obtain an accurate localization. Finally, the Dual-LSTM network is constructed to predict the localization and build an electronic fence to guarantee the safety of the vehicle, guaranteeing the updating of short-distance localization information of the vehicle when the above-mentioned fusion localization is unreliable.

Different from existing methods, this paper acquired accuracy and reliable localization without adding other sensors or using other signals. Furthermore, the proposed method has a low cost for computing capacity, which means the method can be applied to the experimental platform with low computational capacity. The platform can provide more computing usage for other algorithms, such as lidar point cloud fusion, local path planning, etc. Lower computing usage for the in-vehicle platform means a lower risk of a communication jam or algorithm error.

Vehicle platform tests were performed to compare the proposed method with SLAM localization and EKF localization. The results reveal that the proposed method can accurately judge the localization drift between GPS and laser SLAM and achieve accurate positioning in complex and changeable scenes. Compared with the EKF fusion positioning method, the root mean square error is reduced by 66%, reaching centimeter-level positioning accuracy and providing reliable support for autonomous vehicle decision planning in complex scenarios. However, the proposed method cannot solve the localization problem totally; the method can improve the robustness and accuracy of the localization module in severe environments, but to really solve the problem, further study is still needed.

## Figures and Tables

**Figure 1 sensors-24-04025-f001:**
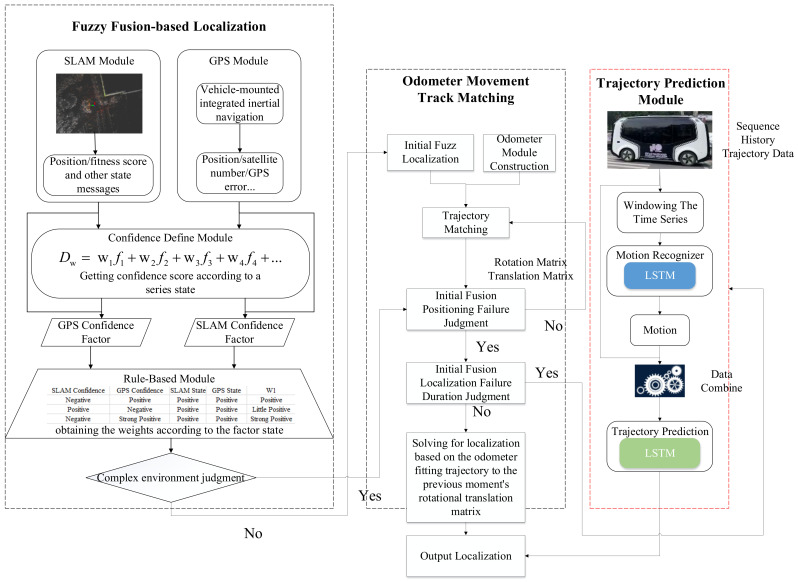
Localization system architecture.

**Figure 2 sensors-24-04025-f002:**
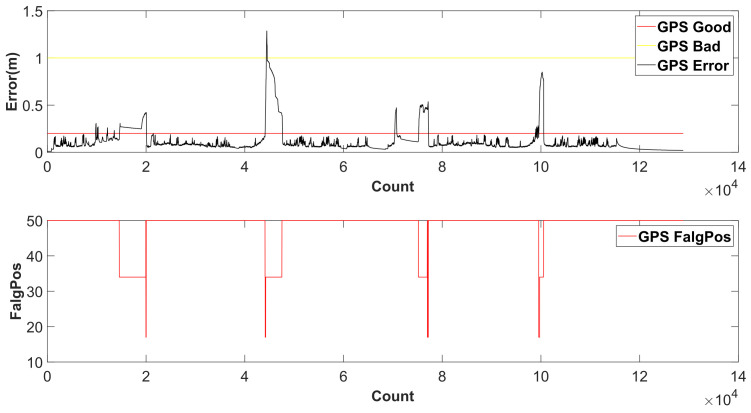
GPS outputs in complex environment.

**Figure 3 sensors-24-04025-f003:**
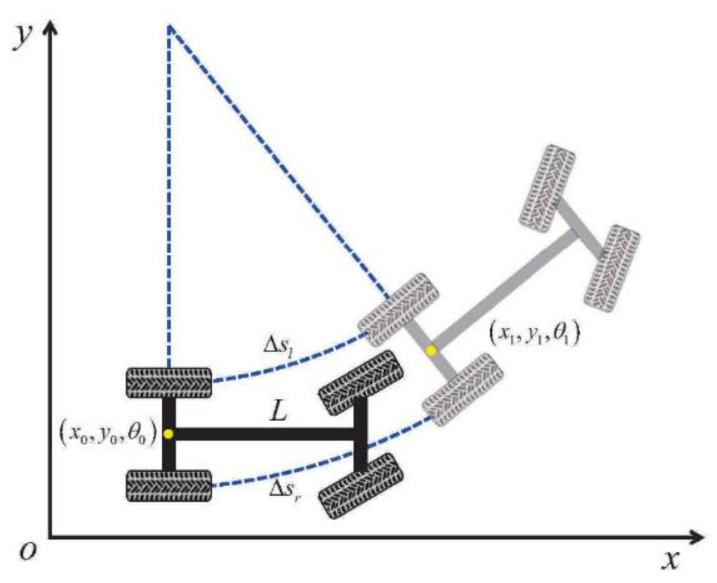
Schematic diagram of odometry calculation.

**Figure 4 sensors-24-04025-f004:**
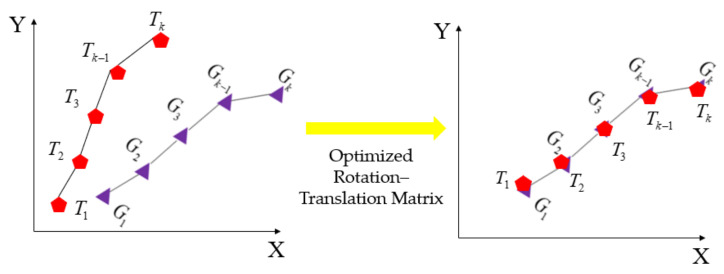
Schematic diagram of trajectory matching.

**Figure 5 sensors-24-04025-f005:**
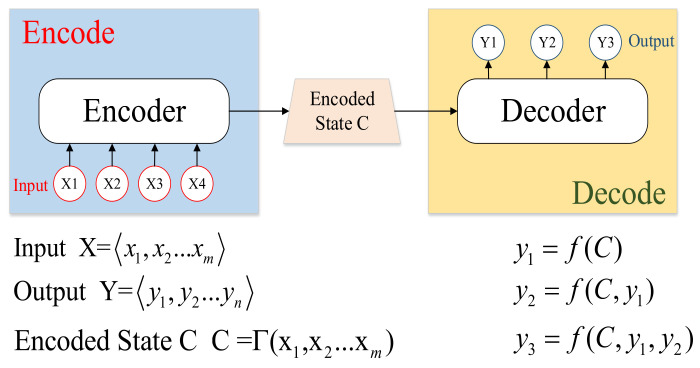
Architecture of the encode–decode mode.

**Figure 6 sensors-24-04025-f006:**
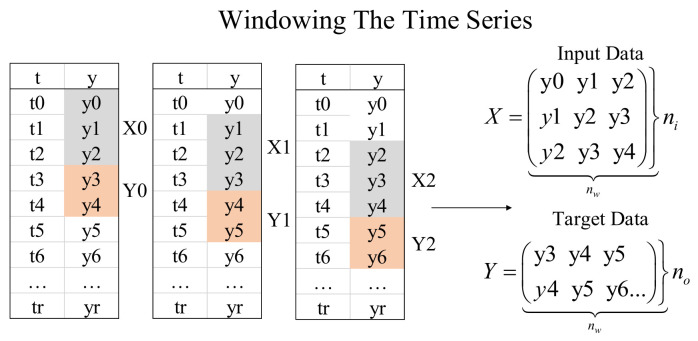
Efficacy explanation of data processing.

**Figure 7 sensors-24-04025-f007:**
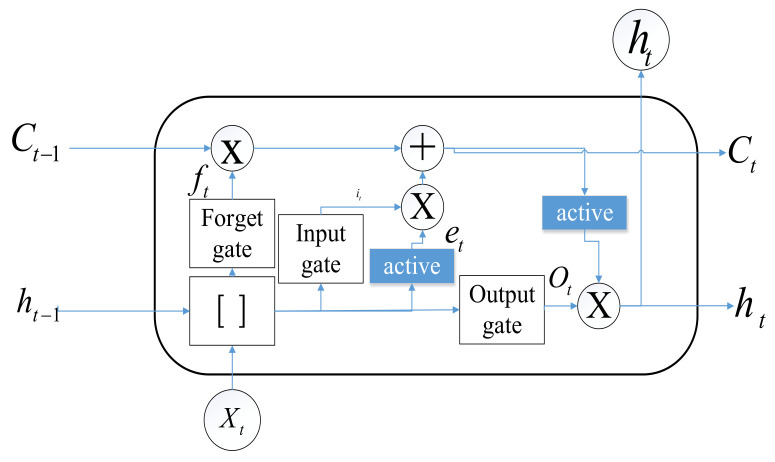
LSTM internal structure.

**Figure 8 sensors-24-04025-f008:**
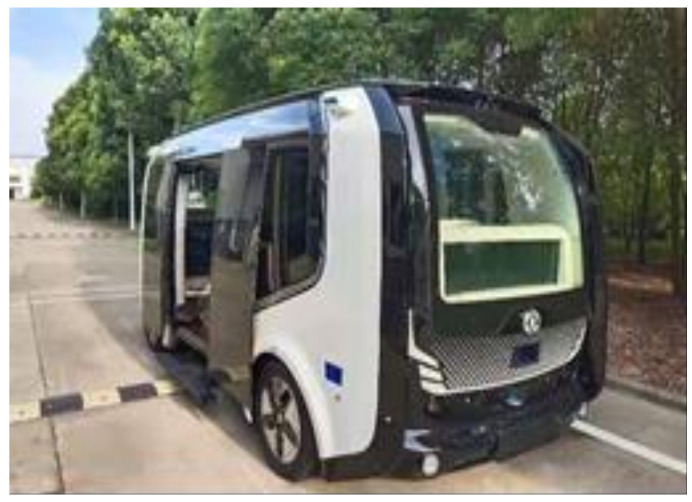
SV 1.0 plus vehicle platform.

**Figure 9 sensors-24-04025-f009:**
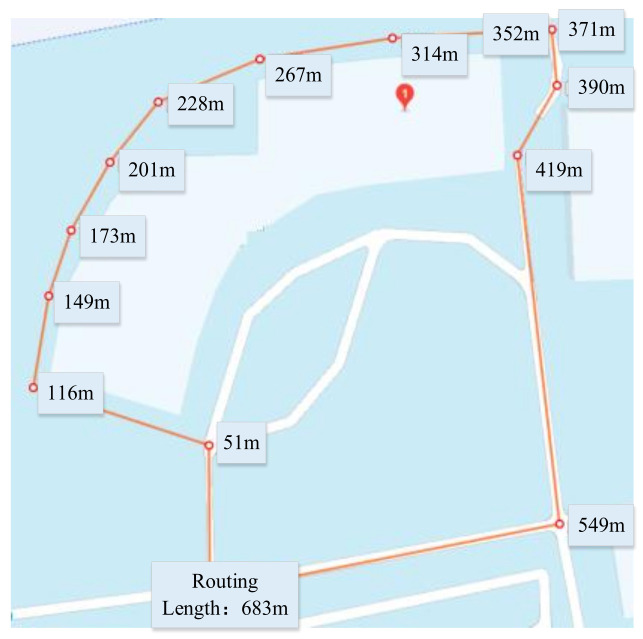
Routing map.

**Figure 10 sensors-24-04025-f010:**
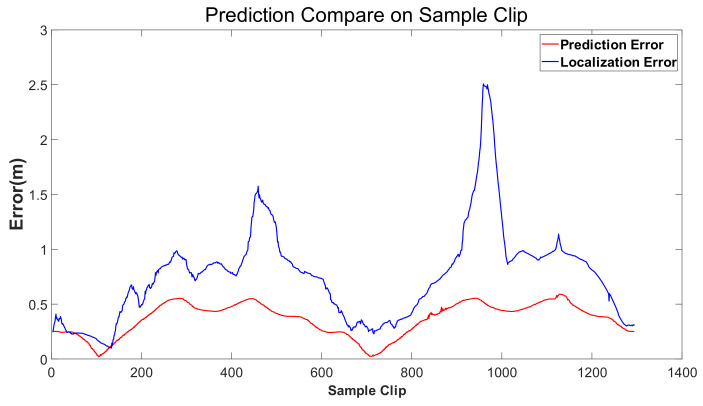
The network prediction result.

**Figure 11 sensors-24-04025-f011:**
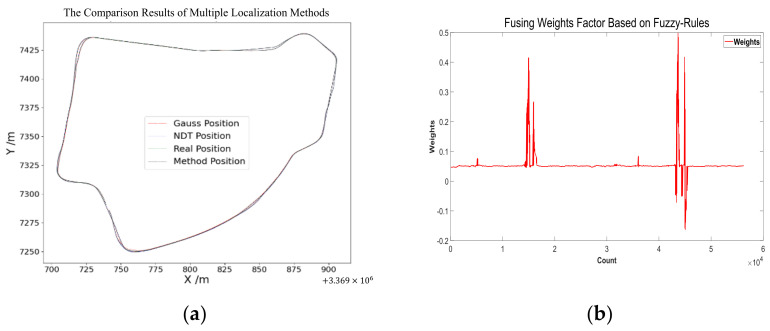
The final test of the module.

**Figure 12 sensors-24-04025-f012:**
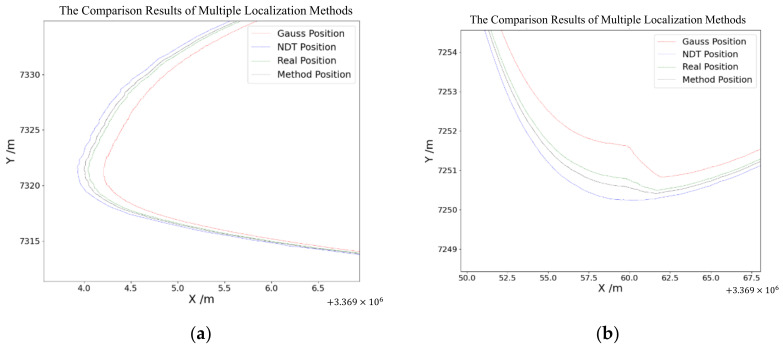
The detailed comparison results of the method.

**Figure 13 sensors-24-04025-f013:**
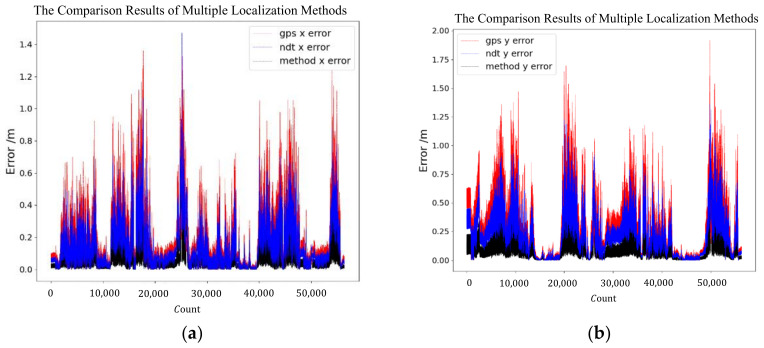
The test error of three localization methods.

**Figure 14 sensors-24-04025-f014:**
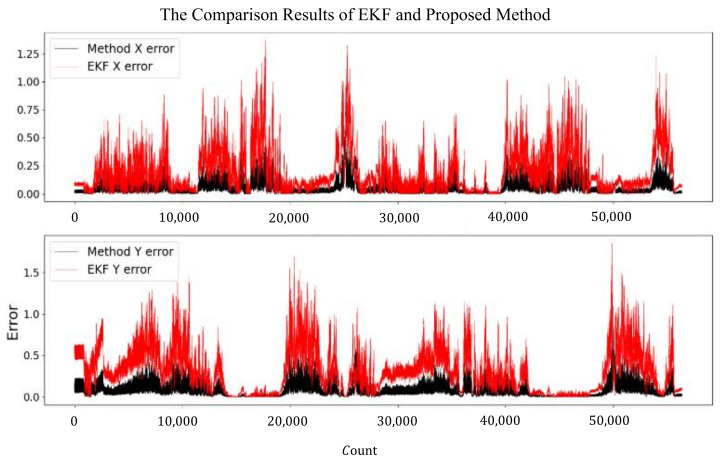
The test error of two localization methods.

**Table 1 sensors-24-04025-t001:** Part of the defined rules.

SLAM Confidence Factor	GPS Confidence Factor	SLAM States	GPS States	W1
Negative	Positive	Positive	Positive	Positive
Positive	Negative	Positive	Positive	Little Positive
Negative	Strong Negative	Positive	Positive	Strong Positive

**Table 2 sensors-24-04025-t002:** Comparison of localization errors in real vehicle test.

Localization Method	MAX Error (m)	RMSE (m)
GPS	1.55	0.505
NDT	1.23	0.338
EKF	1.12	0.433
Proposed Method	0.78	0.144

## Data Availability

The data are not publicly available due to restrictions of company, their containing information that could compromise the privacy of research participants.

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
