# Peer review of "An Integrated LSTM-Rule-Based Fusion Method for the Localization of Intelligent Vehicles in a Complex Environment"

_sensors, 2024, doi:10.3390/s24124025_

Round 1
Reviewer 1 Report
Comments and Suggestions for Authors
- The 'conclusion' section seems to be misplaced under 'IV. Experiments'. It might be more appropriate to place it under section V. This will help maintain the logical flow of your paper.
- The paper mentions that experiments were conducted to test the proposed method. However, the design of the experiment is not clearly presented. Could you provide more details on the variables of interest, blocking factors, control groups, etc.? This will help readers understand the significance of your results.
- The authors' choice of algorithms, particularly LSTM, is not clearly justified. Could you explain why LSTM was chosen over other approaches like GRUs or ResNet? This will enhance the credibility of your methodology. Additionally, it might be beneficial to include tests using other approaches in your experiment.
- The literature review is too shallow. For a topic with many developments in the field, it is reasonable to expect that the paper could provide a comprehensive literature review. Could you reorganize your literature review using a thematic approach, highlighting essential themes? It might help if you first start providing your readers with what intelligent vehicles are and why the localization of such vehicles is a significant problem that must be addressed. In this way, you can show the significance of your proposed approach.
- How did you evaluate that the localization of one method was better than the other? Apart from the histograms, I do not see any table or statistics showing how one was better. Was the judgment purely qualitative? This should be discussed under a data treatment section. Without a proper design of the experiment, we won't be able to rely on the claims you've made in your paper regarding the performance of the proposed approach.
The authors also did not clearly elaborate on the paper's contribution to the literature, so the novelty of the paper is not obvious from the current version.
- A lot of statements are incomprehensible. You may need to proofread your paper and improve the language.
Comments on the Quality of English LanguageI strongly suggest that the authors thoroughly check their manuscript. A lot of statements are incomprehensible.
Author Response
Thank you. Following your comment, we have revised the paper and added description in the file.

Reviewer 2 Report
Comments and Suggestions for Authors
The main motivation of this manuscript (briefly explained in the Introduction section), shows us the work carried out that mainly consists of three modules: the localization module based on fuzzy fusion, the odometer motion tracking matching module, and the trajectory prediction. (it is the use of fuzzy rule theory in the design of the localization module, also using relatively simple NNAs instead of RL, where it is guaranteed that the computational capacity of the vehicle controller can satisfy the localization deviation." (page 6)). Great job, just a few comments:
- The way of citing the terms within the work has to be more careful, since it is not done. example: Eqn(), Figure(), Table().
- The editing of the formulas and tables must be corrected, they are confusing and blurry (Figure 3 is suggested to be made in table form).
- The same treatment is suggested for images since sharpness is compromised.
- Present the vehicle platform more descriptively (Cinematics, Dynamics).
- It is suggested to present characteristics of the dataset data and different rates (apart from the 50 Hz used), in a dynamic table for a better understanding of the behavior of the system.
- In the existing literature there is a large number of works related to the design of control methods or techniques under the so-called “measurement errors”. The authors' conclusion is correct. They must clarify the differences between these approaches to the reader and conclude by clarifying "However, the proposed method cannot completely solve the localization problem, the method can improve the robustness and accuracy of the localization module in environments severe, but to really solve the problem, it still needs more study. ”, this is due to the search for low computational consumption by using simple NNA and low-resource sensors.
Comments on the Quality of English LanguageMinor editing of English language required
Author Response
Thank you. Following your comment, we have revised the paper and added the description in the file.

Round 2
Reviewer 1 Report
Comments and Suggestions for Authors
Most of my comments were addressed. However, upon reading the manuscript, I found some errors, such as typographical errors, that need to be addressed. For example, there are repeated words in a single sentence (see Line 452 in the revised manuscript).
Comments on the Quality of English LanguageThe authors may need to have their paper proofread by a professional editing service. Some of the contents could be made much clearer.
Author Response
Thank you. We appreciate the valuable comments from the Reviewer. We have revised the paper, some of the contents have been made much clearer. The detail response are shown in the attach file.
